# Predictors of Self-Assessed and Actual Knowledge about Diabetes among Nursing Students in Saudi Arabia

**DOI:** 10.3390/jpm13010057

**Published:** 2022-12-27

**Authors:** Abdulellah M. Alsolais, Junel Bryan Bajet, Nahed Alquwez, Khalaf Aied Alotaibi, Ahmed Mansour Almansour, Farhan Alshammari, Jonas Preposi Cruz, Jazi Shaydied Alotaibi

**Affiliations:** 1Department of Nursing, College of Applied Medical Sciences, Dawadmi Campus, Shaqra University, Sahqra 11451, Saudi Arabia; 2Department of Nursing, College of Applied Medical Sciences, Majmaah University, Al-Majmaah 11952, Saudi Arabia; 3Medical Surgical Department, College of Nursing, University of Hail, Hail 81481, Saudi Arabia; 4Department of Medicine, School of Medicine, Nazarbayev University, Astana City 010000, Kazakhstan

**Keywords:** diabetes, diabetes knowledge, nursing students, Saudi Arabia

## Abstract

The aim of this study was to investigate the predictors of self-assessed and actual knowledge of diabetes among undergraduate nursing students. Nursing education plays an important role in preparing future nurses and ensuring that they are knowledgeable and competent in diabetes care. A descriptive, cross-sectional study was conducted with a convenience sample of 330 undergraduate Saudi student nurses. We collected data from October to December 2019 using the Diabetes Self-report Tool (DSRT) and Diabetes Basic Knowledge Tool (DBKT). We performed a multiple regression analysis to identify the predictors of self-reported and actual knowledge of diabetes. The students’ overall mean (SD) scores in the DSRT and DBKT were 48.31 (5.71), which is equivalent to 80.52% of the total score and 22.54 (8.57), respectively. The students’ university, gender, year level and experience in providing direct care to diabetic patients were the significant predictors of self-reported knowledge, whereas their university, age and perceived diabetes knowledge were the significant predictors of actual diabetes knowledge. The findings underscore the necessity to improve student nurses’ actual knowledge of diabetes and its management. Our findings provide a solid basis for planning and implementing educational interventions with diabetes-related information to ensure adequate diabetes knowledge among nursing students.

## 1. Introduction

Diabetes is one of the world’s most common medical problems and its morbidity is increasing at a disturbing rate [1]. The prevalence of diabetes worldwide is anticipated to increase from 463 million in 2019 to 700 million in the 2045 [2]. In the Kingdom of Saudi Arabia, the incidence rate of the disease is high and has become a leading public health problem [1,3]. According to the current investigation by the Saudi Scientific Diabetes Society, Saudi Arabia is among the top 10 nations in the world with the highest prevalence of diabetes [4]. Approximately 17.9% of Saudi adults have diabetes and it is a strong possibility that many more are undiagnosed or pre-diabetic, leaving many on the edge of the disease [4]. This is a foremost concern due to its high prevalence, but the government and health experts are implementing measures to address the lifestyle choices that lead to diabetes and promote healthy ways of living.

Several issues and challenges related to the delivery of quality care for diabetes patients have been identified in the literature. For example, a study conducted in Ethiopia reported poor glycaemic control and poor blood glucose testing practices [5]. Other issues identified were poor adherence to medication, diet and lifestyle modifications [6,7]. Educating patients about diabetes-related information, including its pathology, management, complications and patient self-care, is a critical aspect in improving the quality of diabetes care. However, the literature indicates that patients receive inadequate instructions and training regarding diabetes-related information, which are essential in enhancing knowledge, modifying behaviour, increasing compliance to a treatment regimen and self-care and increasing quality of life [8]. Provider-related factors such as being unprepared to provide diabetes education to patients, inadequate training or education related to diabetes, low priority given to patient teaching and lack of time to conduct patient teaching were identified as barriers in providing quality diabetes education to patients [8].

As one of the largest populations in health care and having the most frequent interactions with patients, nurses play a critical role in diabetes education. However, this dynamic role and responsibility in diabetes awareness are often not performed properly by nurses [8]. Nurses acknowledge that ignorance, the large number of patients in their care and self-doubt are some of the causes for this deficiency. Several research investigations have been undertaken to test the knowledge of nurses in various areas related to diabetes and nursing care for patients with diabetes. A study from the Republic of Rwanda reported a low knowledge level among nurses in terms of diabetes education, specifically on diet, complications, insulin use and impact of stress [9]. A previous review indicated the substantial knowledge inadequacies of nurses from developed and developing nations in the core areas of diabetes care, including ‘insulin therapy, oral diabetes medications, nutrition, BGM, diabetes complications and foot care, diabetes pathology, symptoms and management’ [10].

The significance of education and training in improving nurses’ knowledge regarding diabetes and its care has been emphasized in previous scholarly works [8,10]. The foundation of nurses’ knowledge must start in the undergraduate nursing program. Nursing students are the future of the nursing profession; hence, nursing education plays an important role in preparing future nurses and ensuring that they are knowledgeable and competent in diabetes care. Moreover, nursing students are involved in providing direct care to patients with diabetes during their clinical rotations. However, a previous study argued that nursing students had a low level of awareness of the disorder and that adequate education on the development and management of the disease is necessary to develop their understanding about the disease and to assist in the management of problems associated with diabetes [11]. Hence, nursing students must be equipped with adequate knowledge regarding this topic. Research studies addressing the understanding of diabetes among nursing students are scarce and inadequate. Therefore, it is imperative to investigate student nurses’ level of knowledge of diabetes and to identify areas that need attention to ensure a high level of diabetes knowledge. The aim of the present study was to investigate the predictors of self-assessed and actual knowledge of diabetes-related information in undergraduate student nurses.

## 2. Materials and Methods

### 2.1. Study Design

This study used a quantitative, cross-sectional design.

### 2.2. Setting and Sampling

Undergraduate student nurses enrolled in Shaqra University and Hail University were surveyed in this research study. Shaqra University is located in Riyadh province, while Hail University is located in Hail province. In this study, the convenience sampling method was used for sample selection. The Baccalaureate of Science in Nursing (BSN) program in the two universities is a 4-year program, with 1 year of intensive clinical internship. The inclusion criteria for this study were second-year, third-year, fourth year and intern nursing students of both sexes who were officially enrolled at the time of the investigation. The exclusion criteria were non-nursing students and first-year nursing students (university preparation). The latter were excluded because they did not have major nursing subjects yet. A post hoc power analysis (G*Power v3.1) of the present sample size was performed. The analysis revealed that the sample size achieved 100% power at a medium effect size and 5% significance level (*n* = 330). The questionnaires were distributed to 645 nursing students at the college, of whom 330 completed and returned them, indicating a 51.16% response rate. The study had no missing data, so the final sample included 330 nursing students.

### 2.3. Ethical Considerations

Owing to the absence of an institutional review board in the university during the conduct of the study, the study was approved by the Scientific Research Unit of the College of Applied Medical Sciences, Shaqra University, Saudi Arabia, on 6 October 2019 (approval No. RU-0016). The Scientific Research Unit was responsible for ensuring that the conduct of research activities in the college strictly adheres to the rules governing the ethics of scientific research by the university. The students were given the information they needed about the research and those who decided to participate in the study affixed their signatures to the informed consent form. The researchers explained to the participants that their participation is voluntary and will not affect their performance at the university in any way.

### 2.4. Instrument

The questionnaire was divided into three parts. Part one was designed to elicit data on demographic variables, including the students’ university, age, sex, year level and experience in providing direct care to diabetes patients; the courses, workshops and conferences on diabetes assessment or management that they had attended; and their perceived competence in diabetes nursing care.

The second part was the Diabetes Self-report Tool (DSRT) adapted from Drass et al., (1989) [12], which has 15 items that measure respondents’ self-reported knowledge of diabetes and diabetes care. The items asked about respondents’ knowledge of diabetes-related information such as diabetes pathology, signs and symptoms, assessments, management, nutrition and complications. Four response options were graded according to the following: 1, ‘strongly disagree’; 2, ‘disagree’; 3, ‘agree’; and 4, ‘strongly agree’. The total score was computed, ranging from 15 to 60, to obtain the students’ perceived knowledge. According to the authors, high scores denoted high levels of perceived knowledge [12].

The third part of the questionnaire was the Diabetes Basic Knowledge Tool (DBKT), also by Drass et al., (1989) [12], which measures respondents’ actual knowledge. This part contained 49 multiple-choice questions. The respondents were asked to choose the correct answer from the options provided. Five conceptual dimensions were measured using this tool: (1) diabetes pathology, symptoms and management (14 items); (2) blood glucose monitoring (6 items); (3) diabetes medication (16 items); (4) diabetes diet/nutrition (6 items); and (5) diabetes foot care and complications (7 items). The possible range of scores was from 0 to 49, with higher scores denoting higher levels of actual knowledge. The researchers were granted permission by the original authors to adapt the tool in the study. The two tools distributed to the respondents were in the English language, as in the original version. The validities and reliabilities of the DSRT and DBKT were previously reported and found to be acceptable [13]. When used in Saudi Arabia, the entire questionnaire had a content validity of 0.98 and excellent stability reliability (DSRT: *r* = 0.835, *p* < 0.01; DBKT: *r* = 0.727, *p* < 0.01) [14]. In our sample, Cronbach’s alpha was 0.688 for the DSRT. For the DBKT, a Kuder-Richardson 20 test was performed and an alpha value of 0.887 was obtained. These results attest to the reliability of both the DSRT and DBKT when used among Saudi nursing students [15].

### 2.5. Data Collection

Data were collected from October to December 2019. The respondents were given a maximum of 1 h to answer the questionnaire. To ensure that their actual knowledge was measured accurately, we collected the data in a quiet and comfortable classroom. The students were not allowed to access their books, notes, or the internet or discuss the questions with their classmates. After the allocated time, we collected the questionnaires and kept them inside a secure cabinet until the end of the data gathering period.

### 2.6. Statistical Analysis

The data were analysed using SPSS version 22.0. Descriptive analyses were performed on the demographic variables, self-reports and actual knowledge. Confidence intervals (CIs) were also calculated as necessary. To identify the significant predictors of self-reported and actual diabetes knowledge (dependent variables), multiple regression models were performed on each dependent variable. The demographic variables were used as predictors. For actual knowledge, the students’ perceived knowledge was also included as a predictor variable to examine their association, considering a *p* value of ≤0.05 to be significant.

## 3. Results

A total of 400 questionnaires were distributed, of which 330 were retrieved and entered for analysis (response rate, 82.5%). Other questionnaires were not included in the analysis because of incomplete information and answers. The students’ mean (SD) age was 23.22 (3.85) years (range, 19–36 years). Most students were studying in University B (65.5%; *n* = 216), were female (57.0%; *n* = 188), had experience in giving direct care to patients with diabetes (87.0%; *n* = 284) and had not attended courses, workshops, or conferences on diabetes and diabetes care (59.4%; *n* = 196). The fourth-year students comprised the highest proportion (29.4%; *n* = 97) of the sample, whereas the second-year students comprised the lowest proportion (20.9%; *n* = 69). More than a third of the respondents reported a fair level of competence in diabetes care (43.0%; *n* = 142), 27.6% (*n* = 91) reported good competence, 22.1% (*n* = 73) reported excellent competence and 7.3% (*n* = 24) reported poor competence (see Table 1).

### 3.1. Self-Reported Diabetes Knowledge and Its Predictors

The overall mean (SD) score was 48.31 (5.71) (range, 15–60), which is equivalent to 80.52% of the total score. In the DSRT, item 3, ‘I can identify the long-term complications associated with diabetes,’ garnered the highest mean (mean (SD), 3.53 [0.77]), followed by item 4, ‘I can explain/describe the action and effect of insulin’ (3.52 (0.76)) and item 8, ‘I am generally comfortable teaching patients about insulin therapy’ (3.37 (0.76)). The lowest mean was recorded for item 13, ‘I can list the steps of insulin administering’ (mean [SD], 2.99 (1.20)). Most students reported their agreement with all items on perceived knowledge (Table 2).

To identify the significant predictors of perceived knowledge, the demographic variables of the respondents and their overall perceived knowledge scores were entered as predictor and dependent variables, respectively, into a regression analysis. The model was significant (*F*
_[9, 320]_ = 23.72, *p* < 0.001), accounting for 38.3% of the variance of self-reported diabetes knowledge (*R*^2^ = 0.400, adjusted *R*^2^ = 0.383). The students’ university, sex, year level and experience in providing direct care to diabetic patients were recognized as significant predictors of self-reported knowledge. As indicated in Table 3, the students from University B had greater perceived knowledge than those from University A (*β* = 7.38, *p* < 0.001; 95% CI, 6.02–8.74). Being female was associated with higher levels of perceived knowledge (*β* = 1.16, *p* = 0.025; 95% CI, 0.15–2.17). The third-year students reported higher levels of diabetes knowledge than the second-year students (*β* = 1.79, *p* = 0.031; 95% CI, 0.17–3.41). The students who had prior experience in providing direct care to patients with diabetes reported higher levels of diabetes knowledge than those who did not have a similar experience (*β* = 1.85, *p* = 0.026; 95% CI, 0.22–3.47).

### 3.2. Actual Diabetes Knowledge and Its Predictors

Table 4 summarises the findings of the descriptive analyses of the students’ actual diabetes knowledge from the DBKT. The overall mean (SD) score of the students was 22.54 (8.57), with scores ranging from 1 to 34. The mean (SD) scores for the dimensions were as follows: diabetes pathology, symptoms and management (6.02 [2.33]); blood glucose monitoring (2.29 [1.09]); diabetes foot care and complications (3.85 [1.95]); diabetes diet/nutrition (2.27 [1.10]); and diabetes medication (8.11 [3.65]).

The regression model was statistically significant (*F*
_[10, 319]_ = 142.29, *p* < 0.001), explaining an 81.1% variance in actual diabetes knowledge (*R*^2^ = 0.817, adjusted *R*^2^ = 0.811). The university attended, age and perceived diabetes knowledge were significant predictors of actual diabetes knowledge. Specifically, the students from University B attained scores higher by 15.80 points than those of the students from University A (*p* < 0.001; 95% CI, 14.48–17.11). A year increase in the respondents’ age resulted in a 0.15-point decrease in their actual knowledge score (*p* = 0.022; 95% CI, −0.28 to −0.02). Finally, a 1-point increase in self-reported diabetes knowledge score resulted in a 0.11-point increase in actual diabetes knowledge score (*p* = 0.020; 95% CI, 0.02–0.20; Table 5).

## 4. Discussion

This investigation examined the self-reported and actual knowledge of diabetes and their predictors among Saudi undergraduate nursing students. The findings showed good perceived knowledge, as indicated by the mean score of 48.31 from a possible total score of 60. Surprisingly, the present finding is higher than the perceived diabetes knowledge of nurses in the same country [14]. According to Alotaibi et al. nurses’ mean (SD) score in the same tool was 46.9 (6.1), or 78.2%. The same study reported that nurses overrated their knowledge compared with their actual knowledge [14]. This may also hold true in the present sample. Most student nurses agreed on all statements in the scale. This implies that the students thought that their knowledge in relation to diabetes-related information. The students reported that their lowest level of knowledge was on how to administer insulin and on the proper diet for patients with diabetes and self-care management. These three topics are vital in teaching patients with diabetes. One responsibility of nurses is to teach diabetic patients how to self-administer insulin, the type of diet appropriate for optimum functioning and how to take care of themselves [16]. Hence, additional education related to these topics is needed to ensure that nursing students acquire the right knowledge to teach their patients effectively.

The poor diabetes knowledge of the students in our study is similar to the knowledge reported previously among nursing students in Jordan [17]. In contrast, significantly higher levels of diabetes knowledge were reported among nursing students from Japan and Australia [18]. The barriers to obtaining high levels of diabetes knowledge include a lack of education and training on diabetes pathology and management and a lack of access to educational materials [14].

Int this study, the students had more knowledge on information related to diabetic foot care and other diabetic-related complications and diabetes medications. The poorest knowledge was related to diet or nutrition, followed by blood glucose monitoring and information related to the pathology, symptoms and management of diabetes. Poor knowledge on nutrition and the pathology, symptoms and management of diabetes was also evident among the nurses in Saudi Arabia [14]. The results show gaps in the students’ self-reported and actual knowledge. For example, the students perceived that their knowledge in identifying the long-term complications of diabetes was excellent, but their actual knowledge of the complications of diabetes was very poor. Another example was in relation to blood glucose monitoring, where approximately 87.9% of the students agreed that they were comfortable in instructing patients about glucose monitoring, but their actual knowledge score in this area was very poor.

The good level of perceived knowledge among the students was related to being enrolled in University B. This finding is also true with actual knowledge, where students from University B scored higher than those from the other university. The differences in self-reported and actual diabetes knowledge between the students from the two universities may be related to several factors such as curricular content, teaching methodologies and strategies and the expertise of faculty members [19]. The nursing curriculum and course content in the two universities were not similar. In an earlier study, nursing students from an Australian university showed better clinical diabetes knowledge than nursing students from a Japanese university [20]. The authors justified that these differences in knowledge may be due to the Australian students’ higher frequency of exposure in caring for diabetes patients and knowledge of more people with diabetes [20]. This is also true in the present sample, as indicated by the findings that students who had reported a direct involvement in providing diabetes care had higher perceived diabetes knowledge than those who did not have the same experience. A study involving 52 practice nurses in Australia reported that nurses with ≥2 years of experience exhibited greater knowledge than those with <2 years of experience [21]. In relation to nursing students, clinical practice allows the application of the theoretical learning component of nursing courses and solidifies the knowledge acquired from the classroom by clinical application [22].

Another predictor of perceived knowledge was sex, where women perceived their knowledge to be higher than that of men. Sex was also indicated as a predictor of diabetes knowledge of student nurses in a study conducted in Jordan; however, it was not clear which sex was associated with a higher level of knowledge [17]. In the study by Aloitabi et al. male nurses reported higher levels of perceived diabetes knowledge than female nurses; however, the female nurses achieved significantly higher levels of actual knowledge than male the nurses [14]. Although the reasons for the sex differences in our study are not clear, our findings imply that the female Saudi nursing students may have more confidence in their knowledge than the male students, although this did not translate to actual diabetes knowledge. Furthermore, academic level was also identified as a significant predictor of perceived knowledge, where the third-year students reported better knowledge than the second-year students. The concept of diabetes was taught under a third-year subject (Adult Health Nursing 2) and the third-year students had more clinical exposure, giving them more opportunities to care for clients with diabetes than sophomore students.

Finally, self-reported diabetes knowledge was identified as a predictor of students’ actual knowledge. This means that higher levels of perceived knowledge are linked to greater actual knowledge scores. The findings show that most students who self-rated their knowledge as good achieved high scores in the DBKT. A previous study conducted in Hong Kong also reported a positive association between nurses’ self-reported and actual knowledge [23]. However, despite the positive link between self-reported and actual knowledge, the gap between these two variables remains and previous studies have concluded that more often, nurses are unaware of their insufficient actual knowledge, which could eventually lead to poor care delivered to diabetes patients [14].

## 5. Conclusions

The study assessed the self-reported and actual knowledge on diabetes-related information and examined their predictors among a sample of Saudi undergraduate nursing students. The students in the study rated their perceived knowledge higher than their actual knowledge. The gap between their self-reported and actual knowledge on diabetes was evident. The University B students predicted better self-reported and actual diabetes knowledge. The female students in the third year of nursing school, having experience in giving direct care to diabetes patients, predicted higher levels of self-reported knowledge. The higher levels of self-reported knowledge predicted higher levels of actual knowledge.

### 5.1. Recommendations

The findings underscore the necessity to improve the actual knowledge of undergraduate student nurses about diabetes and its management. The findings provide a solid basis for planning and implementing educational interventions on diabetes-related information to ensure adequate diabetes knowledge among nursing students. Educational content must be structured to include those aspects that received poor knowledge ratings, such as nutrition/diet, blood glucose monitoring and diabetes pathology, symptoms and management. Additional information may also focus on diabetes medication, diabetes foot care and diabetes complications. The individual learning needs of students in the area of diabetes must be assessed and taken into consideration in planning and implementing educational interventions to avoid a knowledge gap between the sexes. There should also be a systematic and intentional implementation of diabetes education in both theoretical and clinical courses to ensure the development of knowledge as students progress in the nursing program. The curriculum of all universities in the country must be examined to identify gaps in their contents and to unify them across the country to ensure the achievement of the same learning outcomes.

### 5.2. Limitations

Although this study provides valuable results regarding nursing students’ knowledge of diabetes, it has several limitations. First, the study design may have implications on how the results of the study could be generalized. Second, the study only surveyed two universities, which might not represent all the universities in the country. Studies in the future should use other research approaches and include more universities. The convenience sampling technique used in this research is also a limitation because it is prone to bias. Thus, future investigations should use random sampling to guarantee the generalizability of results. The predictive variables were only focused on the demographic variables, which limited the discussions on them. Future studies should explore other variables that likely influence nursing students’ knowledge of diabetes.

## Figures and Tables

**Table 1 jpm-13-00057-t001:** Demographic data of the respondents (*n* = 330).

Variable	Mean (SD)	Range
Age, years	23.22 (3.85)	19–36
	*n*	%
University		
University A	114	34.5
University B	216	65.5
Sex		
Male	142	43.0
Female	188	57.0
School level		
2nd year	69	20.9
3rd year	71	21.5
4th year	97	29.4
Internship year	93	28.2
Had experience in providing direct care to diabetes patients		
No	43	13.0
Yes	287	87.0
Attended courses, workshops, or conferences on diabetes and diabetes care		
No	196	59.4
Yes	134	40.6
Perceived competence in diabetes care		
Poor	24	7.3
Fair	142	43.0
Good	91	27.6

**Table 2 jpm-13-00057-t002:** Results of the descriptive analysis of perceived diabetes knowledge (*n* = 330).

Item	Strongly Disagree	Disagree	Agree	Strongly Agree	Mean (SD)
	*n* (%)	*n* (%)	*n* (%)	*n* (%)	
1	21 (6.4)	37 (11.2)	127 (38.5)	145 (43.9)	3.20 (0.88)
2	44 (13.3)	38 (11.5)	101 (30.6)	147 (44.5)	3.06 (1.05)
3	14 (4.2)	14 (4.2)	86 (26.1)	216 (65.5)	3.53 (0.77)
4	12 (3.6)	17 (5.2)	88 (26.7)	213 (64.5)	3.52 (0.76)
5	9 (2.7)	19 (5.8)	248 (75.2)	54 (16.4)	3.05 (0.57)
6	5 (1.5)	25 (7.6)	152 (46.1)	148 (44.8)	3.34 (0.68)
7	10 (3.0)	30 (9.1)	182 (55.2)	108 (32.7)	3.18 (0.71)
8	11 (3.3)	24 (7.3)	128 (38.8)	167 (50.6)	3.37 (0.76)
9	36 (10.9)	21 (6.4)	110 (33.3)	163 (49.4)	3.21 (0.98)
10	13 (3.9)	27 (8.2)	134 (40.6)	156 (47.3)	3.31 (0.79)
11	52 (15.8)	37 (11.2)	90 (27.3)	151 (45.8)	3.03 (1.10)
12	22 (6.7)	44 (13.3)	89 (27.0)	175 (53.0)	3.26 (0.93)
13	71 (21.5)	24 (7.3)	71 (21.5)	164 (49.7)	2.99 (1.20)
14	17 (5.2)	66 (20.0)	100 (30.3)	147 (44.5)	3.14 (0.91)
15	18 (5.5)	72 (21.8)	98 (29.7)	142 (43.0)	3.10 (0.93)
Total score ^a^					48.31 (5.71)

Note. ^a^ Range, 15–60.

**Table 3 jpm-13-00057-t003:** Results of the multiple regression analysis to identify the predictors of perceived knowledge (*n* = 330).

Predictor Variable	*β*	SE-b	Beta	*t*	*p*	95% Confidence Interval	
						Lower	Upper
University	7.38	0.69	0.62	10.70	<0.001 ***	6.02	8.74
Sex	1.16	0.52	0.10	2.26	0.025 *	0.15	2.17
Age	−0.02	0.08	−0.02	−0.29	0.769	−0.18	0.13
Year level (reference group: 2nd year)							
3rd year	1.79	0.82	0.13	2.17	0.031 *	0.17	3.41
4th year	0.86	0.81	0.07	1.07	0.284	−0.72	2.45
Internship year	−0.49	0.85	−0.04	−0.58	0.563	−2.16	1.18
Experience in providing direct care to diabetic patients	1.85	0.83	0.11	2.24	0.026 *	0.22	3.47
Attended courses, workshops, or conferences on diabetes and diabetes care	0.39	0.53	0.03	0.75	0.456	−0.64	1.43
Perceived competence in diabetes care	−0.08	0.32	−0.01	−0.23	0.815	−0.71	0.56

Note. The students’ perceived diabetes knowledge was the dependent variable. *β* is the unstandardized coefficient; SE-b, the standard error; and beta, the standardized coefficient. *R*^2^ = 0.400; adjusted *R*^2^ = 0.383. * Significant at 0.05. *** Significant at 0.001.

**Table 4 jpm-13-00057-t004:** Results of the descriptive analyses of actual diabetes knowledge (*n* = 330).

Variable	Incorrect Answer		Correct Answer		Mean (SD)	Score Range
	*n*	%	*n*	%		
Diabetes pathology, symptoms and management					6.02 (2.33)	0–10
Item 1	145	43.9	185	56.1		
Item 2	246	74.5	84	25.5		
Item 3	291	88.2	39	11.8		
Item 4	151	45.8	179	54.2		
Item 5	123	37.3	207	62.7		
Item 6	252	76.4	78	23.6		
Item 7	134	40.6	196	59.4		
Item 8	127	38.5	203	61.5		
Item 9	139	42.1	191	57.9		
Item 10	299	90.6	31	9.4		
Item 11	142	43.0	188	57.0		
Item 12	166	50.3	164	49.7		
Item 13	303	91.8	27	8.2		
Item 14	115	34.8	215	65.2		
Blood glucose monitoring					2.29 (1.09)	0–5
Item 15	119	36.1	211	63.9		
Item 16	128	38.8	202	61.2		
Item 17	117	35.5	213	64.5		
Item 18	282	85.5	48	14.5		
Item 19	284	86.1	46	13.9		
Item 20	295	89.4	35	10.6		
Diabetes foot care and complications					3.85 (1.95)	0–7
Item 21	155	47.0	175	53.0		
Item 22	135	40.9	195	59.1		
Item 23	290	87.9	40	12.1		
Item 24	110	33.3	220	66.7		
Item 25	125	37.9	205	62.1		
Item 26	130	39.4	200	60.6		
Item 27	95	28.8	235	71.2		
Diabetes diet/nutrition					2.27 (1.10)	0–5
Item 28	268	81.2	62	18.8		
Item 29	103	31.2	227	68.8		
Item 30	136	41.2	194	58.8		
Item 31	119	36.1	211	63.8		
Item 32	312	94.5	18	5.5		
Item 33	294	89.1	36	10.9		
Diabetes medication					8.11 (3.65)	0–14
Item 34	108	32.7	222	67.3		
Item 35	128	38.8	202	61.2		
Item 36	301	91.2	29	8.8		
Item 37	257	77.9	73	22.1		
Item 38	138	41.8	192	58.2		
Item 39	143	43.3	187	56.7		
Item 40	118	35.8	212	64.2		
Item 41	105	31.8	225	68.2		
Item 42	98	29.7	232	70.3		
Item 43	127	38.5	203	61.5		
Item 44	132	40.0	198	60.0		
Item 45	116	35.2	214	64.8		
Item 46	302	91.5	28	8.5		
Item 47	127	38.5	203	61.5		
Item 48	129	39.1	201	60.9		
Item 49	274	83.0	56	17.0		
Total actual diabetes knowledge score					22.54 (8.57)	1–34

**Table 5 jpm-13-00057-t005:** Results of the multiple regression analysis to identify the predictors of actual knowledge (*n* = 330).

Predictor Variable	*β*	SE-b	Beta	*t*	*p*	95% Confidence Interval	
						Lower	Upper
University	15.80	0.67	0.88	23.68	<0.001 ***	14.48	17.11
Sex	0.29	0.43	0.02	0.66	0.509	−0.56	1.13
Age	−0.15	0.07	−0.07	−2.31	0.022 *	−0.28	−0.02
Year level (reference group: 2nd year)							
3rd year	−0.13	0.69	−0.01	−0.18	0.857	−1.48	1.23
4th year	0.75	0.67	0.04	1.12	0.265	−0.57	2.07
Internship year	1.12	0.71	0.06	1.59	0.114	−0.27	2.51
Experienced in providing direct care to diabetic patients	0.18	0.69	0.01	0.26	0.798	−1.18	1.54
Attended courses, workshops, or conferences on diabetes and diabetes care	0.37	0.44	0.02	0.84	0.402	−0.49	1.23
Perceived competence in diabetes care	−0.16	0.27	−0.02	−0.59	0.558	−0.68	0.37
Perceived diabetes knowledge	0.11	0.05	0.07	2.33	0.020 *	0.02	0.20

Note. The students’ actual diabetes knowledge was the dependent variable. *β* is the unstandardized coefficient; SE-b, the standard error; and beta, the standardized coefficient. *R*^2^ = 0.817; adjusted *R*^2^ = 0.811. * Significant at 0.05. *** Significant at 0.001.

## Data Availability

Not applicable.

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
