# Peer review of "Predictors of Self-Assessed and Actual Knowledge about Diabetes among Nursing Students in Saudi Arabia"

_jpm, 2022, doi:10.3390/jpm13010057_

Round 1

Reviewer 1 Report

Thank you for the opportunity to review your manuscript titled Predictors of Self-Assessed and Actual Knowledge Regarding Diabetes among Nursing Students in Saudi Arabia. It's an interesting study, and it highlights the need to improve the competencies of nursing students in the care of diabetic patients.

Author Response

Dear Reviewer,  
I attach document revision for your valuable comments here 

Thank you so much 

Thanks 

Reviewer 2 Report

Alsolais et al. conducted a cross-sectional survey to evaluate nursing students’ self-reported and actual knowledge about diabetes and diabetes care. The findings could contribute to adding validated evidence on the matter, especially the results on the nurses’ overrating knowledge compared to their actual knowledge on the topic. even if the context is extremely local. Further training and education efforts should be devoted to students’ education.

I have some other general and specific comments that could help improve the paper’s quality.

-       Please, consider an English language revision of the manuscript.

-       From a public health perspective, it could be useful adding details, if available, on the knowledge about vaccinations in people with diabetes.

-       I believe the “strengths and limitations” section is completely missing. Discussing the limitations of the study and taking into account sources of potential bias or imprecision is mandatory.

-       Please, provide details on the eligibility criteria of the study population and the selection of the participants, even if the sample was of convenience.

-       In the methods section, it could be useful to add details on the efforts to address potential sources of bias and to explain how missing data were addressed.

-       In the results section, please report the number of individuals at each stage of the study and consider using a flow diagram, also indicating the number of participants with missing data for each variable.

-       Please, consider translating beta coefficient estimates into relative risk or odds ratio in order to improve readability for the biomedical public.

- I suggest authors could better put in context the extremely local data. 

Author Response

Dear author

Thank you so much for your valuable comment. Attach my response to your valuable comments with two files. One is the manuscript, and I highlight the fixed comments in red or blue. At the same time, I have attached a one-word file to respond to your comments. I also attach the receipt for the editing services. 

Thanks

Round 2

Reviewer 2 Report

Thank you for addressing comments.